# Annual exposure to PAHs in urban environments linked to wintertime wood-burning episodes

Irini Tsiodra [1,2], Georgios Grivas [3], Kalliopi Tavernaraki [1], Aikaterini Bougiatioti [3], Maria Apostolaki [1], Despina Paraskevopoulou [3,2], Alexandra Gogou [5], Constantine Parinos [5], Konstantina Oikonomou[6], Maria Tsagkaraki [1], Pavlos Zarmpas [1], Athanasios Nenes [2,4*] & Nikolaos Mihalopoulos [1,3*]

[1]Environmental Chemical Processes Laboratory, Department of Chemistry, University of Crete, Heraklion,71003, Greece,

[2] Center for the Study of Air Quality and Climate Change, Institute of Chemical Engineering Sciences, Foundation for Research and Technology Hellas, Patras, GR-26504, Greece

[3] Institute for Environmental Research and Sustainable Development, National Observatory of Athens, Lofos Koufou, P. Penteli, Athens, 15236, Greece

[4] Laboratory of Atmospheric Processes and their Impacts, School of Architecture, Civil & Environmental Engineering, Ecole polytechnique fédérale de Lausanne, Lausanne, CH-1015, Switzerland

[5] Hellenic Centre for Marine Research, Institute of Oceanography, 190 13 Anavyssos, Attiki, Greece

[6] CARE-C Research Center, The Cyprus Institute, Nicosia 2121, Cyprus

*Correspondence to*: Athanasios Nenes (athanasios.nenes@epfl.ch) & Nikolaos Mihalopoulos (nmihalo@noa.gr)

**Abstract.** Polycyclic aromatic hydrocarbons (PAHs) are organic pollutants in fine particulate matter (PM) long known to have mutagenic and carcinogenic effects, but much is unknown about the importance of local and remote sources for PAH levels observed in population-dense urban environments. A year-long sampling campaign in Athens, Greece, where more than 150 samples were analyzed for 31 PAHs and a wide range of chemical markers, was combined with Positive Matrix Factorization (PMF) to constrain the temporal variability, sources and carcinogenic risk associated with PAHs. It was found that biomass burning (BB), a source mostly present during wintertime intense pollution events (observed for 18% of measurement days in 2017), led to wintertime PAH levels 7 times higher than in other seasons and was as important for annual mean PAH concentrations (31%) as diesel/oil (33%) and gasoline (29%) sources. The contribution of non-local sources, although limited on an annual basis (7%), increased during summer, becoming comparable to that of local sources combined. The fraction of PAHs (12 members that were included in the PMF analysis) that was associated with BB was also linked to increased health risk compared to the other sources, accounting for almost half the annual PAH carcinogenic potential (43%). This can result in a large number of excess cancer cases due to BB-related high PM levels and urges immediate action to reduce residential BB emissions in urban areas facing similar issues.

**Keywords:**

PAHs, Biomass Burning, Air Pollution Evens, Source Apportionment, Receptor Modeling, Positive Matrix Factorization, Carcinogenic potential, Risk Assessment

# 1. Introduction

Polycyclic aromatic hydrocarbons (PAHs) are abundant in the atmosphere and can be present in the gas or particle phase (Finlayson-Pitts, 1997; Lin et al., 2015). Ambient PAHs originate mainly from internal combustion engines, stationary sources such as power and industrial plants, residential heating (including biomass burning), cooking, and wildfires (Finlayson-Pitts, 1997; Simoneit, 2002; Bond et al., 2013). Several PAHs are characterized as potential carcinogens and/or mutagens (IARC,2010; Nowakowski et al., 2017). The IARC Group 1 carcinogen Benzo[*a*]pyrene (BaP) is extensively studied among PAHs and is used as a marker for PAH toxicity. The European Union with the directive 2004/107/EC has set an air quality standard for the carcinogenic risk of PAHs. While the directive highlights the necessity of measurement for several PAHs (at least six other members), it lays down an annual target value only for BaP (1 ng m$^{-3}$), which is selected as a marker for the carcinogenic risk of polycyclic aromatic hydrocarbons in ambient air.

The presence of PAHs tends to be highly enhanced in urban environments (Jiang et al., 2014; Mo et al., 2019; Ringuet et al., 2012), such as Athens, Greece. Studies to date in Athens show that PAH concentrations are significantly enhanced during the cold period of the year (Mantis et al., 2005) and linked to a multitude of incomplete combustion sources. Fireplace and woodstove usage for domestic heating in Greece has increased dramatically since the 2010 financial crisis, having considerable impacts on air quality in major urban centers, such as Athens (Paraskevopoulou et al., 2014; Grivas et al., 2018; Theodosi et al., 2018) and Thessaloniki (Saffari et al., 2013), that host half the population of Greece.

It is worth noting that since 2013 several violations of the EU limit value for BaP have been recorded in Greek cities, with concentrations showing a clear wintertime enhancement (NAPMN, 2020). Greek urban centers are not unique in this regard, as residential BB is a major issue for urban air quality throughout Europe. Studies in Central European cities have reported BB contributions to cumulative particle PAH concentrations that even exceed 50% (Li et al., 2018; Masiol et al., 2020; Schnelle-Kreis et al., 2007), while in other Mediterranean urban areas its impact appears less profound (Callén et al., 2014).

Uncontrolled residential BB emissions can lead to the appearance of intense pollution events (IPE), during which very high urban levels of organic aerosols are observed in urban centers worldwide (Florou et al., 2017; Saffari et al., 2013; Fountoukis et al., 2016). Excessive exposure of urban populations to ambient woodsmoke pollution can cause severe health effects (Kocbach Bølling et al., 2009; Naeher et al., 2007; Wong et al., 2019), especially in view of the very high concentrations of highly-oxidized organic species from nighttime chemistry, and their potential to enhance toxicity (Kodros et al., 2020).

Moreover, the contribution of wildfires to regional background PAHs can be significant, given their increased incidence as a result of climate change (McCarty et al., 2020; O'Dell et al., 2020). Especially wildfires near urban centers can have a direct and pronounced effect on population exposure. Every summer since 2017, for example, intense episodes of air quality degradation from wildfire smoke near major Greek urban regions (such as Athens, Patras, Heraklion, Kalamata and Pyrgos) have been recorded. During these events, largely increased PM$_{2.5}$ and PM$_{10}$ levels (reaching a few hundreds of μg m$^{-3}$) that persist for several days (https://panacea-ri.gr; Amiridis et al., 2012; Stavroulas et al., 2019) are observed. Although much work has been done on PAH sources, their attribution using atmospheric samples is often pursued through highly uncertain diagnostic ratios for measured concentrations (Katsoyiannis et al.,

2011). Relatively few studies exist in Europe and N. America that utilize receptor modeling (e.g. Schnelle-Kreis et al., 2007; Sofowote et al., 2008; Iakovides et al., 2021a), an approach that can quantify the relative contributions of distinct local and regional sources to PAH levels and associated carcinogenic risks that urban populations are exposed to.

The present study aims to characterize the variability and sources of PAHs in Athens, Greece, one of the largest urban centers in the Eastern Mediterranean. This is one of the first studies in Greece and Southeastern Europe to utilize PMF receptor modeling for source apportionment of PAHs. For this, filter-based analysis is combined with online chemical speciation, allowing for the association of PAH groups with specific OA components and the verification of PMF-resolved sources. At present, there are only a few studies involving PAHs, that comparatively assess offline and online organic aerosol source apportionment (e.g. Bozzetti et al., 2017; Srivastava et al., 2018). To specifically characterize the diurnal variability of PAHs and the BB source, separate daytime and nighttime filter samples were analyzed here, as done in a some previous studies in Greece (Saffari et al., 2013; Tsapakis and Stephanou, 2007). The separation of diesel and gasoline aerosol sources that is attempted in this work is also highly important for Athens which stands out as a particularity among European cities, given the very small penetration of diesel cars in the passenger fleet due to the ban that existed up to 2011. The resulting identification of carbonaceous aerosol source profiles can provide useful reference for source apportionment studies that use long-term chemical speciation data in urban areas, thus enabling a source-specific toxicity assessment that considers local against regional contributions and is highly important for policies on PAH exposure reduction.

## 2. Materials and Methods

### 2.1 Study area

The Greater Athens Area (GAA, Figure S1), with 3.8 million inhabitants, is one of the largest metropolitan regions in Southern Europe and a major commercial and transportation hub. Nearly 3 million private cars are registered for circulation in the GAA, along with 0.3 million trucks and buses (Hellenic Statistical Authority, 2020). Up to 2011, diesel-powered private cars were banned in Athens and, even though nowadays they comprise the majority of new sales, they only account for about 10% of the fleet. Up to 2011, heating oil was the predominantly used fuel for domestic heating in Athens (in 76% of residences). However, its consumption in the GAA decreased by 61% in 2011-2018 (Hellenic Statistical Authority, 2019), with people resorting to alternative heating sources, including biomass burning in fireplaces and stoves. Industrial activity in the GAA is minor in the central basin while it is mostly concentrated in the Thriassion plain, 10-20 km to the northwest and separated by a low-altitude mountain range. The port of Piraeus, to the southwest, is the largest passenger port in Europe and also one of the busiest container ports in the Mediterranean (Figure S1). The complex topography of the basin favors the appearance of mesoscale flows throughout the year and the frequent stagnation of air masses (Kassomenos et al., 1998).

## 2.2 Sampling and Analyses

Ambient $PM_{2.5}$ samples were collected between December 4, 2016 – January 31, 2018, at the urban background site of the National Observatory of Athens (NOA) at Thissio (37.97326∘ N, 23.71836∘ E), which is located 146m above sea level and is representative of the urban background conditions in central Athens (Stavroulas et al., 2019; Theodosi et al., 2018).

During the non-winter months, two samples per week were analyzed, on alternating days between weeks. This way, a representative distribution between weekday and weekend samples (69% - 31%) was achieved, since it is known that this is an important temporal scale inducing variability in urban PAH levels (Dutton et al., 2010; Lough et al., 2006). The same approach was followed for the winter period, focusing at the same time on as many daytime-nighttime pairs as possible to characterize the diurnal patterns. From December to February, separate daytime and nighttime filters were collected every 12h (6:00-18:00 LST, 18:00-6:00 LST, respectively) for a total of 80 samples, while 76 24-h filters were sampled from March to November. The sampling schedule also satisfies the 2004/107/EC directive requirements for PAHs (roughly 1 measurement per week, equally distributed around the year), allowing the comparison with the BaP target value.

A low-volume sampler (3.1 PNS 15, Comde Derenda GmbH, Stahnsdorf, DE) with a flow rate of 2.3 $m^3$ $h^{-1}$ was used to collect particles onto quartz fiber filters of 47mm diameter (Flex Tissuquartz, Pall Corporation, Port Washington, NY, USA). The inlet height was 2 m above ground. Field and laboratory blanks were routinely collected. PAHs in the filter samples were quantified by gas chromatography-mass spectrometry (GC-MS, Agilent 6890N, Agilent Technologies Inc., Santa Clara, CA, USA). Prior to the analysis, samples were spiked with a mixture of deuterated internal standards for identification of PAHs and calculation of recovery efficiencies (16 members, LGC Standards, Middlesex, UK). Extraction was performed following the pre-established procedure of Gogou et al. (1998) with modifications (Parinos et al., 2019) (Supplement, Section S1, Table S1). Briefly, extracts were obtained using a 50:50 n-hexane-dichloromethane mixture and were purified on a silica column. PAHs were eluted with a 10 mL n-hexane/ethyl acetate (9:1 v/v) mixture and placed into a glass vial for further concentration under a gentle nitrogen stream. [$^2H_{12}$]perylene, used as an internal standard, was spiked into the vial before sealing and storage. Limits of Detection (LODs) were calculated as 3 times the standard deviation of blanks. On the day of the analysis, injections with internal standards were also run to calculate relative response factors (RRF). The identification of compounds was based on retention time, mass fractionation and co-injection of standard mixtures.

Organic and elemental carbon (OC, EC) were determined by the thermal-optical transmission (TOT) method, using a Sunset carbon analyzer (Sunset Laboratory Inc., Portland, OR, USA) (Cavalli et al., 2010; Paraskevopoulou et al., 2014). Water-soluble ions were detected by ion chromatography (IC) (Paraskevopoulou et al., 2014) and monosaccharide anhydrides (levoglucosan, mannosan and galactosan) by High-Performance Anion Exchange Chromatography with Pulsed Amperometric Detection (HPAEC-PAD) (Iinuma et al., 2009).

The submicron organic aerosol (OA) fraction was monitored at the same site by an Aerosol Chemical Speciation Monitor (ACSM, Aerodyne Inc., Billerica, MA, USA) (Ng et al., 2011). The ACSM samples ambient air, through a critical orifice to a particle-focusing aerodynamic lens. The particle beam is driven

in the detection chamber where non-refractory particles are flash-vaporized at 600°C and ionized by electron impact. Finally, the resulting fragment ions are characterized by a quadrupole mass spectrometer. The obtained organic mass spectra are decomposed into OA components representative of specific sources-processes (namely hydrocarbon-like OA, biomass burning OA, cooking OA, semi-volatile and low-volatility oxygenated OA), through Positive Matrix Factorization (PMF), based on the multilinear engine ME-2 algorithm and using the Source Finder toolkit (SoFi) (Canonaco et al., 2013; Bougiatioti et al., 2014). The standard sampling duration of the Q-ACSM that was used is 30 min. ACSM monitoring was carried out for the majority of the filter-sampling period (Dec-2016 to Jan-2018). Data on OA, ionic composition and organic spectra were available with the exception of September and October 2017. Validated, PMF-resolved OA source contributions were available during December 2016 – July 2017. Additional details on the OA measurement and analysis can be found in Stavroulas et al., (2019).

Equivalent Black carbon (BC) levels were monitored by a 7-wavelenth aethalometer (AE33, Magee Scientific, Berkeley, CA, USA), with compensation for loading (DualSpot) and multiple scattering effects (Liakakou et al., 2020). The instrument applies internally the two-component Aethalometer model (Drinovec et al., 2015; Sandradewi et al., 2008) and provides online estimates of the BB (%) contribution to BC at 880 nm, using the assumption that absorbing aerosols from fossil fuel (FF) combustion and biomass burning (BB) have distinct spectral properties.

Regulatory pollutants and meteorological parameters (temperature, wind speed-direction) were routinely measured at Thissio. Reference-grade instrumentation was used for measurement of CO (NDIR, APMA 360, Horiba Inc.), $NO_x$ (chemilluminescence, APNA 360 Horiba Inc.) and $O_3$ (UV-Vis, 49-i, Thermo Fisher Scientific Inc.). $NO_x$ concentration data were also obtained from a nearby regulatory (Greek Ministry of Environment and Energy) traffic site (Athinas Str., 0.9 km to NE of Thissio). ACSM, Aethalometer, gaseous pollutant, temperature and wind speed data were averaged on an hourly basis.

Four-day air mass back trajectories, arriving at Thissio at 1000m (Grivas et al., 2018; Kalkavouras et al., 2020; Stavroulas et al., 2019), were calculated using the Hybrid Single Particle Lagrangian Integrated Trajectory (HYSPLIT) model and grouped daily using cluster analysis (Kalkavouras et al., 2020). The altitude of 1000m was chosen to capture the regional transport with a high probability of affecting pollution within the boundary layer (hence, air quality and population exposure) and is consistent with typical planetary boundary layer estimates over Athens (Kokkalis et al., 2020).

**2.3 Source apportionment**

PMF analysis was performed using the EPA PMF 5.0 model (Norris et al., 2014). The 12-h samples collected during the winter intensive campaigns were appropriately combined to 24-h periods, between 6:00 LST of each day and 6:00 LST of the following. In total, 104 24-h samples were considered for the analysis. The carbonaceous aerosol speciation dataset that was used consisted of OC and EC, PAHs, oxalate and levoglucosan. Total carbon (TC: sum of OC, EC) was included as a total variable in the PMF model, for a meaningful mass balance approach (Piletic et al., 2013; Valotto et al., 2017). In total, 16 species were entered in the analysis, including 12 PAH members. Rotational and random errors were assessed using the bootstrap (BS) and displacement (DISP) error estimation methods included in the EPA

PMF 5.0 software (Paatero et al., 2014). Details on PMF model design parameters, solution metrics, uncertainties and error indices are provided in the Supplement (Section S2, Table S2).

**2.4 PAH contribution to carcinogenic risk**

The "toxic equivalence factor" (TEF) approach was used to estimate the carcinogenic potency of measured PAHs, in which the toxicity of each member is expressed using BaP as reference (Taghvaee et al., 2018a):

$$\sum BaP_{eq} = \sum (C_i \; TEF_i) \tag{1}$$

where $C_i$ is the concentration (ng m$^{-3}$) and $TEF_i$ is the Toxicity Equivalent Factor of each member (Bari et al., 2010; Nisbet and LaGoy, 1992). The lifetime excess cancer risk (ECR) from inhalation was estimated as follows:

$$ECR = \sum BaP_{eq} \; UR_{BaP} \tag{2}$$

where $UR_{BaP}$ (unit risk) refers to the number of excess cancer cases in the population with chronic inhalation exposure to 1 ng m$^{-3}$ of BaP over a lifetime of 70 years. Estimations were made using reference $UR_{BaP}$ values of $1.1 \times 10^{-6}$ (0.11 cases per 100,000 people) – according to the Office of Environmental Health Hazards Assessment (OEHHA) of the California Environmental Protection Agency (CalEPA) – and $8.7 \times 10^{-5}$ (8.7 cases per 100,000 people) – according to the World Health Organization (WHO) (Alves et al., 2017; Elzein et al., 2019).

# 3. Results and Discussion

## 3.1 Temporal Variability

Figure 1 presents the monthly variation of total PAH concentrations, displaying a notable wintertime enhancement as compared to the summer period. Mean ΣPAH concentrations in winter 2016-2017, spring 2017, summer 2017, autumn 2017 and winter 2017-2018 were $7.0 \pm 10.1$, $2.5 \pm 4.1$, $0.9 \pm 0.9$, $2.2 \pm 3.2$ and $22.3 \pm 26.8$ ng m-3, respectively, indicating a clear seasonal cycle. The mean annual (2017) BaP concentration in this study (0.26 ng m$^{-3}$) did not exceed the EU target value, but nevertheless was higher than the WHO reference level (0.12 ng m$^{-3}$). Largely increased BaP levels were observed during wintertime episodic events. These results can be placed in context by comparison against past studies for PAHs in the GAA (Table S3) that identified similar seasonal profiles, although not with such pronounced winter-summer differences. Moreover, this study reports the highest mean annual BaP concentrations at a background location in the GAA in over two decades (Marino et al., 2000). In the few studies in the area that have compared traffic with background sites, there appears to be an important roadside enhancement of PAH levels (Andreou and Rapsomanikis, 2009; Mantis et al., 2005; Pateraki et al., 2019). Therefore, it is noteworthy that the present, urban background, BaP annual mean concentration is comparable to the mean BaP concentrations reported at 21 sites in the GAA by a study of annual duration in 2010-2011, 7 of which were high-traffic locations. Specifically, the street-site concentrations were reportedly higher by 44 and 55% for ΣPAHs and BaP, respectively, compared to urban background sites

(Jedynska et al., 2014). This comparison likely indicates an increase of urban background levels in Athens, with implications for the population's exposure.

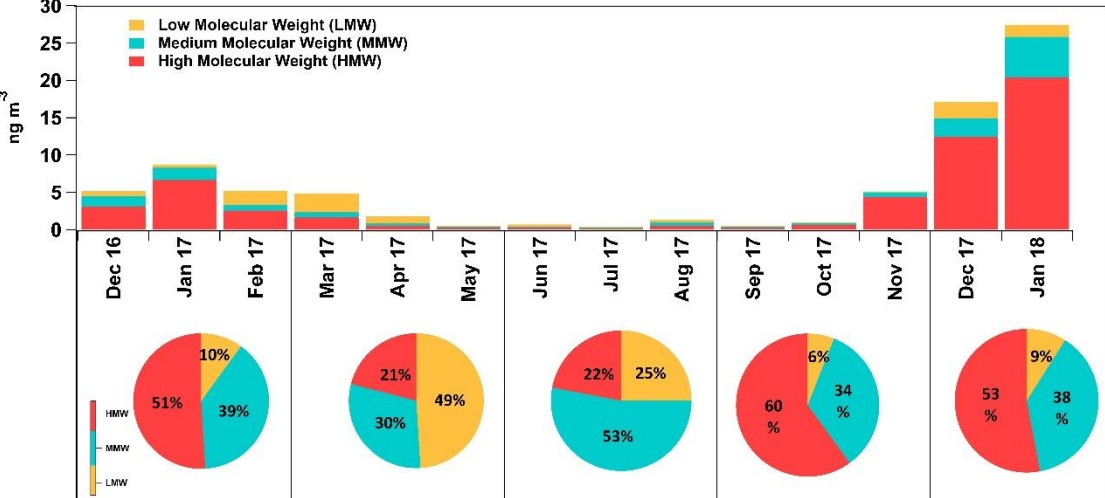

**Figure 1: The color bars represent the mean monthly concentrations of high (HMW), medium (MHW) and low (LMW) molecular weight PAHs, between December 2016 – January 2018. The pie-charts in the lower panel show the concentration fractions of LMW, MMW and HMW PAHs in each season.**

The classification of PAHs by molecular weight is thought to provide information about their sources. For example, low molecular weight PAHs (LMW; 128-178 gmol[-1]) and medium molecular weight PAHs (MMW; 202-228gmol[-1]) have been linked to diesel engine emissions (Zheng et al., 2017), while MMW and high molecular weight (HMW; 252-300 gmol[-1]) PAH concentrations can be influenced by BB emissions (Han et al., 2019; Masiol et al., 2020). Figure 1 displays also the ΣPAH fractions of LMW, MMW and HMW PAHs, by-season of the year. During winter and autumn, HMW PAHs comprised the most abundant fraction (60% and 51%, respectively), implying a possible impact from BB. In spring there was an increase of the LMW fraction (49%), while during summer, when regional fine aerosol sources maximize their influence (Grivas et al., 2018), the higher percentage (53%) came from MMW species.

A prior study carried out during 2001-2002 (Table S3), well-before the economic recession in Greece, at a GAA site not adjacent to a major road (Mantis et al., 2005), reported an LMW-MMW-HMW PAH fractionation similar with the present study (Figure S2), despite the considerably higher residential BB emissions here. However, the present concentrations of ΣPAHs and BaP were higher, by a factor of 1.4 and 1.8, respectively. This most likely suggests a significant effect of the new BB source on an annual level, since PAH emissions from the road transport sector in Greece remained relatively comparable between the two periods (EIONET, 2021). These results indicate that approaches based on MW alone have limitations, since many PAH members are emitted by both fossil fuel and biomass burning, and the attribution of PAH sources should be addressed through receptor modelling.

The seasonal variability seen in Figure 1 is assessed statistically (t-tests) to identify significant seasonal differences (winter against non-winter measurements), with results presented in Table S4. Higher mean

PAH concentrations were observed during wintertime, when emissions from processes like biomass burning are more intense. Lower PAH levels were recorded during the non-winter period due to the additional effects of volatilization, photochemical degradation and enhanced atmospheric dispersion. For all PAH members, concentrations were significantly higher during the winter season at the 0.95 confidence level with the exception of Σ C1-202 (which however registered a large number of BDL values).

Temperature-dependent processes like gas/particle partitioning can affect the observed seasonal variability of semi-volatile PAH compounds. Previous studies in Greece (Iakovides et al., 2021b; Mandalakis et al., 2002; Sitaras and Siskos, 2001, 2008; Terzi and Samara, 2004; Vasilakos et al., 2007), that performed both gas- and particle-phase measurements, reported that LMW PAHs were found predominantly in the gas phase (gas-phase fractions: 88-97%), MMW PAHs in both gas and particle phase (gas phase: 46-90%) and HMW PAHs mainly in the particle phase (particle phase: 71-100%). In the present study, only the particle phase of PAHs was measured. In an effort to assess the fractionation of PAH members between the particle phase (measured) and the gas phase in the dataset, an approximation was performed, based on the partitioning theory of semi-volatile organics due to absorption by organic material in the particle phase (Pankow, 1994). This approach, and the partitioning-absorption equations used to calculate the partitioning coefficient $K_p$ and therefore the gas/particle concentration ratios, have been used by numerous PAH studies (e.g. Andreou and Rapsomanikis, 2009; Xie et al., 2013). The estimations here were found to be in agreement with the above mentioned studies in Athens and indicate that volatilization leads to minimal fractions of particle LMW PAHs (found in the gas-phase in mean fractions of 71-100%). Regarding MMW, Flt and Pyr are the main compounds affected by volatilization (15-25%, respectively). HMW members remain relatively unaffected, being partitioned in the particle phase in fractions of 91-100%, even though during regional transport and through successive dilutions, a substantial part of HMW could end up being lost from the particle phase due to physical volatilization. The partitioning characterization methodology and results are presented in section S3 of the Supplement (Figure S3). Finally, as it has been proposed, some organic components can adopt an amorphous solid or semisolid state, that can further impact the rate of heterogeneous reactions and multiphase processes (Shiraiwa et al., 2011). In our study for PAHs this does not seem to be the case, as PAHs from outside the region of Athens will be shown later in the paper to be present at low levels.

Apart from volatilization processes, the temporal variability of emission sources can account for a major part of observed seasonal differences. Correlations of PAHs with primary (NO$_x$, CO) and secondary (O$_3$) pollutants can provide insights into the related sources and atmospheric processes. The correlation analysis was performed separately for winter and non-winter months and results are presented in section S3 of the Supplement (Table S5). LMW members and Flt, Pyr showed weak correlations with primary pollutants (NO$_x$, CO), that were slightly higher during the cold period. MMW and HMW members were highly correlated with CO at Thissio and NO$_x$ at the traffic site during both the winter and non-winter periods. For those members, the highest wintertime correlations ($r$: 0.62-0.88, $p < 0.01$) signify the additional contribution of biomass burning and heating oil combustion emissions that co-emit heavier PAHs and CO, NO$_x$. In the non-winter period there were relatively higher correlations of Cor, IP and

BghiP with CO and NO$_x$ as measured at Thissio ($r$: 0.55-0.81, $p < 0.01$), that probably indicate an impact from light-duty vehicles (Weiss et al., 2011), in the absence of domestic burning.

**3.2 Investigating intense pollution events: nighttime vs. daytime**

Summary statistics for 12- and 24-h ΣPAH averages are provided in Table 1. The highest PAH concentrations were observed during intense pollution events (IPE), that in this work are defined as periods with mean BC concentrations exceeding 2 μg m$^{-3}$, stagnant conditions with wind speeds below 3 ms$^{-1}$ and a lack of precipitation (Fourtziou et al., 2017). Peak wintertime levels of BC (Figure S4)

coincided with BC$_{bb}$ maxima, indicating a predominance of biomass burning over other local combustion sources for carbonaceous aerosols during these events (Liakakou et al., 2020). These high levels are considered to be mainly driven by emissions and not changes in mixing-layer height (Liakakou et al., 2020).

**Table 1: Average values ± standard deviation of PAH concentrations categorized according to their molecular weight, during wintertime intense pollution events (IPE).**

|  | LMW (128-178 gmol$^{-1}$) (ng m$^{-3}$) | MMW (202-228 gmol$^{-1}$) (ng m$^{-3}$) | HMW (252-300 gmol$^{-1}$) (ng m$^{-3}$) | ΣPAHs (ng m$^{-3}$) |
|---|---|---|---|---|
| IPE DAY | 1.79 ± 1.95 | 1.13 ± 0.50 | 4.01 ± 4.11 | 6.94 ± 4.27 |
| IPE NIGHT | 1.51 ± 1.28 | 4.97 ± 7.69 | 21.56 ± 20.31 | 28.04 ± 27.56 |
| IPE 24-h | 1.62 ± 1.57 | 3.44 ± 6.22 | 14.54 ± 18.05 | 19.6 ± 23.78 |
| Non-IPE DAY | 1.21 ± 1.27 | 0.92 ± 0.77 | 1.81 ± 1.59 | 3.93 ± 2.51 |
| Non-IPE NIGHT | 0.46 ± 0.25 | 0.71 ± 0.48 | 1.61 ± 2.24 | 2.78 ± 2.32 |
| Non-IPE 24-h | 0.92 ± 1.06 | 0.84 ± 0.67 | 1.73 ± 1.83 | 3.49 ± 2.47 |

Among the 80 winter samples, 50 met the above criteria for IPE (20 daytime and 30 nighttime samples). IPEs occurred with higher frequency during the second winter (24 vs. 17, in the common months:

December-January), due to more intense emissions (e.g. the BC mean concentration was 2.6 times higher during the second winter) and reduced atmospheric dispersion (average wind speeds of 3.0 and 1.9 m s$^{-1}$, in the first and second winter periods, respectively).

During IPEs, ΣPAH concentrations displayed a strong night-day gradient, owing to the effect of nighttime residential BB emissions. Several studies have examined the diurnal variability of PAH levels,

worldwide (e.g. Yan et al., 2009). Similarly pronounced nighttime enhancements have been mostly reported for megacities in E. Asia, such as Beijing and Nanjing (Elzein et al., 2019; Haque et al., 2019), where PAH levels are considerably higher than in Athens, due to the intensity and diversity of local sources that include burning of biomass, coal and plastics. During non-IPE periods, ΣPAH levels were lower by 43% in daytime and 90% in nighttime, resulting in daytime levels being higher than nighttime

and indicative of substantial contributions from other combustion sources like vehicular traffic.

HMW PAHs had the highest contribution to ΣPAHs during IPEs and presented the most pronounced day-night differences. MMW PAHs were more abundant during nighttime, with 24-h levels more than

double during IPEs. Levels of LMW PAHs were higher during daytime in both IPE and non-IPE periods, suggesting that biomass burning had a more limited effect in their particle-phase concentrations, compared to the higher MW members.

To explore the various sources that drive the variability of PAHs, the correlations of ΣPAHs with source-specific tracers were examined (Table S6). Given the very high correlations of ΣPAHs with BaP (Figure S5), especially during nighttime IPEs ($r = 0.95$, $p < 0.01$), similar associations with examined tracers can be expected for the corresponding PAH carcinogenicity. Strong correlations with biomass burning tracers such as non-sea salt $K^+$ and $BC_{bb}$ were observed, particularly during IPE nights ($r$: 0.81 and 0.95, respectively, $p < 0.01$). Similarly, levoglucosan, mannosan and galactosan displayed very high correlations with ΣPAHs during nighttime IPEs ($r$: 0.90-0.93, $p < 0.01$). Comparisons were also made with the "fingerprint" ACSM $m/z$ 60 and 73 that are linked to dissociation of levoglucosan produced as a pyrolytic product of biomass (Alfarra et al., 2007; Heringa et al., 2011; Weimer et al., 2008). In fact, filter-based levoglucosan measurements correlated excellently with $m/z$ 60 and 73 ACSM fragments ($r$ = 0.98 and 0.97, respectively, $p < 0.01$). ΣPAHs exhibited strong correlations with $m/z$ 60 and 73 during nighttime IPEs, with $r$ values of 0.88, 0.86 ($p < 0.01$), respectively, indicating the impact of BB emissions. A close association was observed also between ΣPAHs and submicron organic aerosol (OA). During the period when PMF-resolved ACSM OA components were available, the strongest correlations were observed with BBOA and SV-OOA, the components that the cold period in Athens correspond to fresh and processed BB organic aerosol, respectively (Stavroulas et al., 2019). On the contrary, ΣPAHs were uncorrelated with LV-OOA, which mostly represents OA of regional origin. An absence of correlation between ΣPAHs and highly-oxidized OA from ASCM data has been also reported by field and experimental studies (Cui et al., 2020; Zheng et al., 2020).

The results indicate a significant impact of BB emissions on PAH levels during nighttime IPEs. The ratio of ΣPAHs to levoglucosan during these events was 53% lower than in daytime, when biomass burning and therefore levoglucosan concentrations decrease. This indicates the important effects of additional PAH sources such as vehicular traffic, a finding consistent with past studies in Greece (Saffari et al., 2013). The levoglucosan/(mannosan+galactosan) ratio can also indicate whether solid fuels used for heating are either "aged" (e.g. aged wood or lignite) producing more levoglucosan, or more "fresh" (Saffari et al., 2013). The calculated value close to 6 for both daytime and nighttime samples suggests that solid fuels used for residential heating in Athens are mainly associated with fresh firewood and their type has not changed since 2013, when a similar wintertime ratio was reported in Athens (Fourtziou et al., 2017). Finally, the ratio of levoglucosan to mannosan can be indicative of wood type, with hardwood use (e.g., olive, oak, beech) producing ratios around 14–15, while softwood (e.g., pine) burning leads to lower ratios, around 4 (Schmidl et al., 2008). In the present study, the levoglucosan/mannosan ratio ranged from 7.3 to 9.7 (Table S7), indicating a balanced use of both wood types, in agreement with observations at the same site during the 2013-2014 winter (Fourtziou et al., 2017), once again supporting that a mixture of fresh softwood and hardwood is consistently used for residential heating over the recent years.

### 3.3 PMF modeling and source characterization

Solutions with 3-8 factors were examined, with the four-factor solution deemed as the most physically meaningful (a detailed presentation of the selected solution can be found in Section S5 of the supplement). The four identified factors are presented below; a more extensive discussion on their validation is provided in Section S5 of the Supplement.

The first factor was attributed to biomass burning (BB). Levoglucosan, a well-established BB signature marker, is almost exclusively associated with this factor (Figure 2). The factor is also characterized by important loadings in 5-6 ring PAHs, a feature that has been reported in BB source profiles of PMF studies in urban areas worldwide (Han et al., 2019; Masiol et al., 2020; Taghvaee et al., 2018b). The strong presence of BaP in the factor is consistent with results from studies in European cities (Li et al., 2018; Srivastava et al., 2018) and also with a recent study at Thissio (Fourtziou et al., 2017) that reported direct associations between wintertime BaP levels and several BB tracers. The OC/EC ratio in the profile was 4.2, higher than the other local sources and characteristic of fresh BB emissions. It was also comparable to the value (3.7) calculated for the BB source at the same site by a long-term $PM_{2.5}$ source apportionment study (Theodosi et al., 2018). Moreover, the source contributions registered the highest, among the four factors, correlations with BB external tracers and specifically with $BC_{bb}$ ($r = 0.93$, $p < 0.01$) and water-soluble $K^+$ ($r = 0.61$, $p < 0.01$). Similarly, the strongest correlations were observed between the BB source and the $m/z$ 60 and 73 ACSM fragments ($r = 0.88$ and 0.86, respectively, $p < 0.01$), further validating its identification. The biomass burning source was closely related to the BBOA component ($r = 0.85$, $p < 0.01$) with a slope of 0.003 indicating that the 12 members included in the PMF analysis comprise about 0.3% of BBOA (Poulain et al., 2011). Similar associations have been reported by studies comparing OA components from online aerosol mass spectrometry and source contributions from offline filter-based source apportionment (Bozzetti et al., 2017; Srivastava et al., 2021). The bivariate polar wind plot (calculatated according to the methodology of Uria-Tellaetxe and Carslaw, 2014) for contributions of the BB source (Figure S6a) indicates the local character of the source, with concentrations being enhanced during low-wind conditions, as it has been observed at the same site for fresh BB aerosols emitted in central Athens (Kaskaoutis et al., 2021; Stavroulas et al., 2019). The factor was present almost exclusively during the winter months (Figure S7), when local wood-burning emissions for residential heating intensify, leading to frequent IPE.

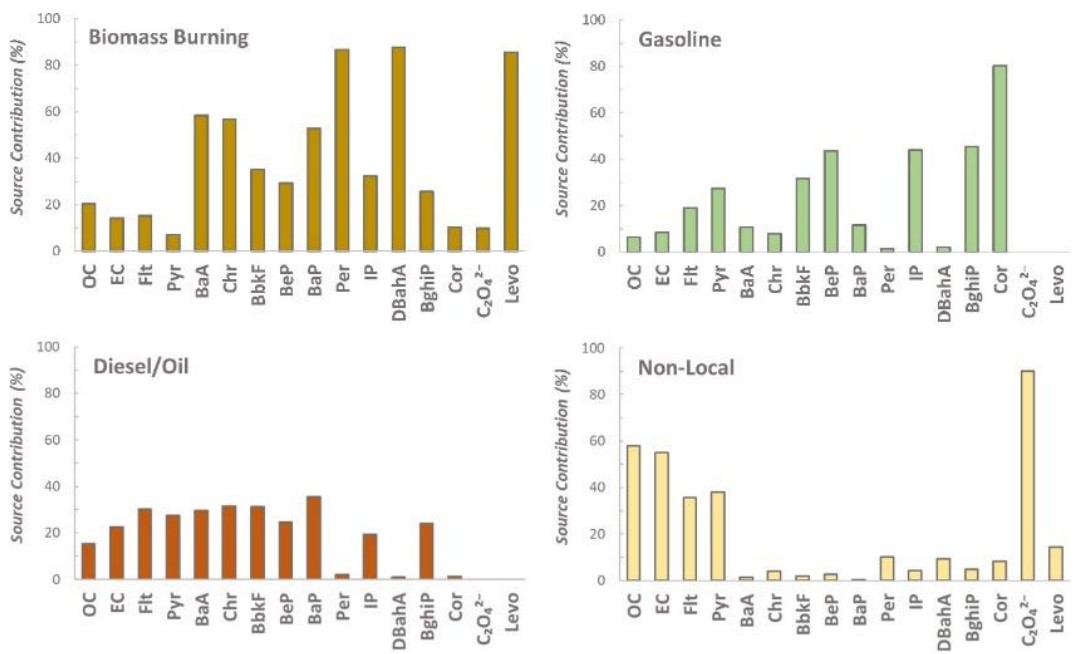

**Figure 2: Percentage contributions of individual sources to mean modelled concentrations of PAHs and other carbonaceous species, obtained from PMF analysis for 24-h samples collected at Thissio Athens, during December 2016 – January 2018.**

The second factor was linked to gasoline-powered vehicles, which constitute the major fraction (over 90%) of the vehicular fleet in the GAA. The factor was characterized by stronger contributions to HMW PAHs (e.g., IP, BghiP and Cor), over lower-MW members (Figure 2). This feature has been used to differentiate the gasoline source from general traffic sources in several PAH source apportionment studies (Callén et al., 2013; Javed et al., 2019; Sofowote et al., 2008; Wu et al., 2014). The OC/EC ratio

in the source profile was 1.9, within the range (1.7-2.3) typically reported for fresh emissions from gasoline vehicles (Grivas et al., 2012). The source contribution time series presented the highest correlations with $BC_{ff}$ ($r = 0.79$, $p < 0.01$), which is mostly a proxy of traffic emissions impacting Thissio (Liakakou et al., 2020). It was the only factor that correlated significantly during both non-winter ($r = 0.80$, $p < 0.01$) and winter ($r = 0.66$, $p < 0.01$) periods with CO, which is emitted mainly by gasoline

vehicles in the GAA (Fameli and Assimakopoulos, 2016). On the contrary the BB source correlated with CO only in winter ($r = 0.83$, $p < 0.01$) due to strong CO emissions from wood burning during IPEs (Gratsea et al., 2017). Moreover, this was also the only local source that was correlated ($r = 0.57$, $p < 0.01$) with ACSM HOA during the non-winter months, when HOA is exclusively associated with vehicular traffic (Stavroulas et al., 2019). However, unraveling the HOA contributions from gasoline and

diesel vehicles is challenging (Shah et al., 2018) and while some experimental results support a close connection of ambient HOA profiles with gasoline vehicle emissions (Collier et al., 2015) there are field source apportionment studies that report high correlations of HOA with both fuel-type combustion sources (Al-Naiema et al., 2018). The wind plot for the factor contributions (Figure S6b), given the absence of a strong directional pattern, is again suggestive of aerosols produced in the vicinity of the site

(central Athens). Higher contributions were observed during the winter months (Figure S7), consistent

with the increased traffic in the center of Athens (especially during the December holiday period), relative to the vacation months of July-August.

The third factor was associated with emissions from diesel/oil combustion and is characterized by an increased abundance of lower MW members (Shirmohammadi et al., 2016; Zheng et al., 2017). It presented the highest contributions to Flt and Pyr among local sources and also substantial loadings in BaA, Chr, BbkF and BaP, along with smaller – compared to the gasoline factor – loadings in IP and BghiP (Figure 2). Comparable patterns can be observed in source profiles of other PMF studies that distinguish diesel and gasoline sources (e.g. Agudelo-Castañeda and Teixeira, 2014; Liu et al., 2019), where contributions in the diesel factor were found relatively increased for 4-, 5-ring PAHs and decreased for HMW members. The OC/EC ratio was 1.6, higher than typically reported values for diesel exhaust, which could indicate moderate aging. Higher OC/EC ratios can be expected also in the cases of heavy-duty diesel vehicles in creeping mode (Pio et al., 2011) and non-traffic oil combustion emissions (e.g. ships in the port). Based on its wind plot, the factor, while still local, presents a relative enhancement for moderate-speed winds transporting aerosols from the S-W sectors of the GAA (Figure S6c). Primary pollution hot-spots are found in this direction, such as the port of Piraeus (at a distance of 7-11 km) and the industrial/commercial hub of the Athens basin (2-4 km) that is traversed by the E75 international route, the most heavily trafficked – and frequently congested – highway in the GAA (Grivas et al., 2019). Therefore, the area to the S-W of the site is characterized by increased circulation of light- and heavy-duty diesel vehicles. As with the gasoline factor, the contribution time-series recorded statistically significant ($p < 0.01$) correlations with external combustion indicators, albeit weaker ($r = 0.66$, $p < 0.01$ with CO; $r = 0.39$, $p < 0.01$ with $BC_{ff}$). Although the factor was uncorrelated with the HOA during the non-winter months, it recorded a large correlation during the first winter period ($r = 0.88$, $p < 0.01$), potentially signifying a measurable contribution of stationary heating fuel combustion – recognized as an additional contributing factor to HOA during winter in Athens (Stavroulas et al., 2019) – to the diesel factor during stagnation events. It is recognized, however, that this correlation could be inflated by a common boundary layer/stagnation effect of traffic and residential emissions and therefore more research is need to confirm the finding. When examining the entire period with availability of OA component data (December 2016 – July 2017), it was observed that the source had a consistent association ($r = 0.74$, $p < 0.01$) with the HOA/CO ratio, that is expected to be higher for aerosols from diesel combustions (Reyes-Villegas et al., 2016).

The fourth factor was characteristic of non-local contributions to carbonaceous aerosol. It was mainly associated with high contributions to OC and EC, which at urban background locations are moderately impacted from local primary sources and mostly driven by aerosols regionally transported to the receptor site (Buzcu-Guven et al., 2007; Hasheminassab et al., 2014). The dominance of regional/secondary sources at urban and suburban background sites has been identified by the majority of aerosol source apportionment studies in the GAA (Diapouli et al., 2017; Grivas et al., 2018; Paraskevopoulou et al., 2015; Theodosi et al., 2018) and other urban areas worldwide, like Mexico City (Aiken et al., 2009) and Paris (Skyllakou et al., 2014). The factor was also characterized by a high contribution of oxalate, an important secondary constituent of water-soluble organic carbon (Myriokefalitakis et al., 2011). The contributions of non-local sources were positively correlated with sulfate and ammonium ($r$: 0.57 and

0.51, respectively, $p < 0.01$), indicators of regionally-transported secondary aerosol. The non-local PMF factor showed the strongest associations with SV-OOA and LV-OOA, the secondary, oxidized OA components (Srivastava et al., 2021; Xu et al., 2021). This was mainly observed during the non-winter months ($r = 0.77$ and 0.55, $p < 0.01$, respectively), while in the winter period the factor was again correlated with LV-OOA of regional origin ($r = 0.48$, $p < 0.05$) but not with SV-OOA, which in winter is rather associated with the fast processing of heating emissions (Stavroulas et al., 2019). This is the only factor that showed a statistically significant ($p < 0.01$) enhancement (Figure S7) during the non-winter months, due to increased photo-oxidation for production of secondary organics under stronger insolation. The polar plot of Figure S6d displays the typically observed (Stavroulas et al., 2019) large dispersion of concentration enhancements along the SW-NE axis of the Athens basin, indicating the association of the factor with transport on a larger-than-urban spatial level. A considerable fraction of Flt and Pyr was attributed to this factor. Studies performed at regional background sites that are mainly impacted by long-range transport have attributed increased contributions of Flt and Pyr to aerosols deriving from distant combustion of coal and heavy oil (Lhotka et al., 2019; Mao et al., 2018; Miura et al., 2019; Wang et al., 2014). The OC/EC ratio in the source profile (2.9) was increased compared to the petroleum-related sources (1.6-1.9), but not as much as usually reported for secondary aerosol factors. Given this, and that PAHs are subject to oxidative aging and removal during atmospheric transport (Galarneau, 2008; Ravindra et al., 2008) it is likely that the non-local factor not only includes transboundary aerosols, but also partially-aged aerosol from a less extended spatial scale (e.g. from energy production using fossil fuels in continental Greece or from emissions from marine oil combustion in the Aegean Sea). In support of this, it is noted that the centroids of all identified air mass trajectory clusters during the study period converge to the north of the GAA before arriving in the Athens basin (see also the discussion in the section 3.4).

### 3.4 Source contributions

Average source contributions to PAHs and TC are presented in Figure 3 for the calendar year 2017 (January to December; 93 24-h values), to achieve consistent statistical sampling of each season. The following discussion is focused on the contributions to PAHs, while the contributions to TC are discussed in Section S5 of the Supplement. The mean annual contributions of sources to mean modeled OC and EC are displayed in Figure S8.

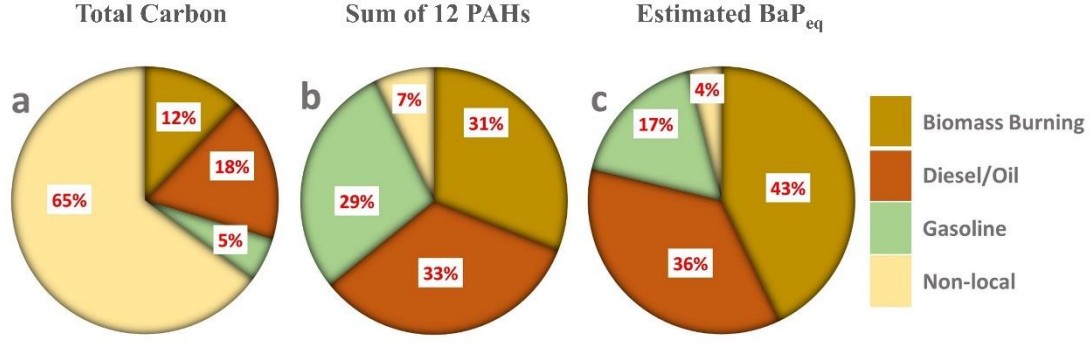

**Figure 3: Fractional contributions of PMF-resolved sources to mean modeled concentrations of: a) TC, b) Sum of 12 PAHs included in the PMF analysis, c) estimated BaP$_{eq}$.**

Regarding local sources, the annual contribution of biomass burning to ΣPAHs is amplified compared to TC (31% vs 12%, Figures 2a, 2b), ever so more when assessing only the contributions to carcinogenic PAHs (36%, Figure 3c). The large impact of biomass burning on long- and short-term PAH exposure becomes more evident considering that it is essentially a source active mainly in wintertime and manifests mostly during IPEs (18% of measurement days in 2017).

The other two local sources (diesel/oil and gasoline) accounted for a combined 62% of ΣPAHs (Figure 3b), highlighting the importance of urban vehicular emissions on a long-term basis – but not during IPEs. Even though the participation of diesel cars is minimal in the passenger fleet in Athens (<10%, the vast majority being Euro 5 and 6 vehicles), the contribution of the diesel/oil factor is at least comparable to the gasoline factor for ΣPAHs (and much larger for TC), indicating that it is probably emissions from older light- and heavy-duty vehicles (and to an extent stationary emissions) that should be associated with the significant contributions of this source. Manoli et al. (2016), reported an even larger gap between diesel and gasoline contributions to ΣPAHs (51% vs 30% approximately), at an urban background location in Thessaloniki, Greece, during 2011-2012. Even before the lifting of the ban on private diesel cars in Athens and Thessaloniki (2011), diesel emissions were considered to be more important regarding the impact of road transport on PAH concentrations (Andreou and Rapsomanikis, 2009; Viras and Siskos, 1993). Comparing with these studies, a declining contribution to PAHs of diesel vehicles in Greek cities is implied.

Non-local sources were the major contributors to total carbon concentrations (65%) at the Thissio site (Figure 3a) – consistent with findings from previous studies at urban background sites in the GAA and elsewhere. The effect of non-local sources is greatly reduced (Figure 3b) when examining contributions to ΣPAHs (7%) against those of the three local sources (29-33%). Figure 4a shows that non-local daily contributions remain lower than 20% during winter but regularly exceed 50% during summer.

Non-local contributions are intimately linked with regional transport; therefore, an air mass trajectory cluster analysis can help understand their variability and origins. Four major air mass source regions are

identified for the GAA during the study period, using 96-h back-trajectories: The Black Sea area (with a
frequency of 43%), Northern Greece/Balkans (32%), Western Europe (20%) and Eastern Europe (5%).

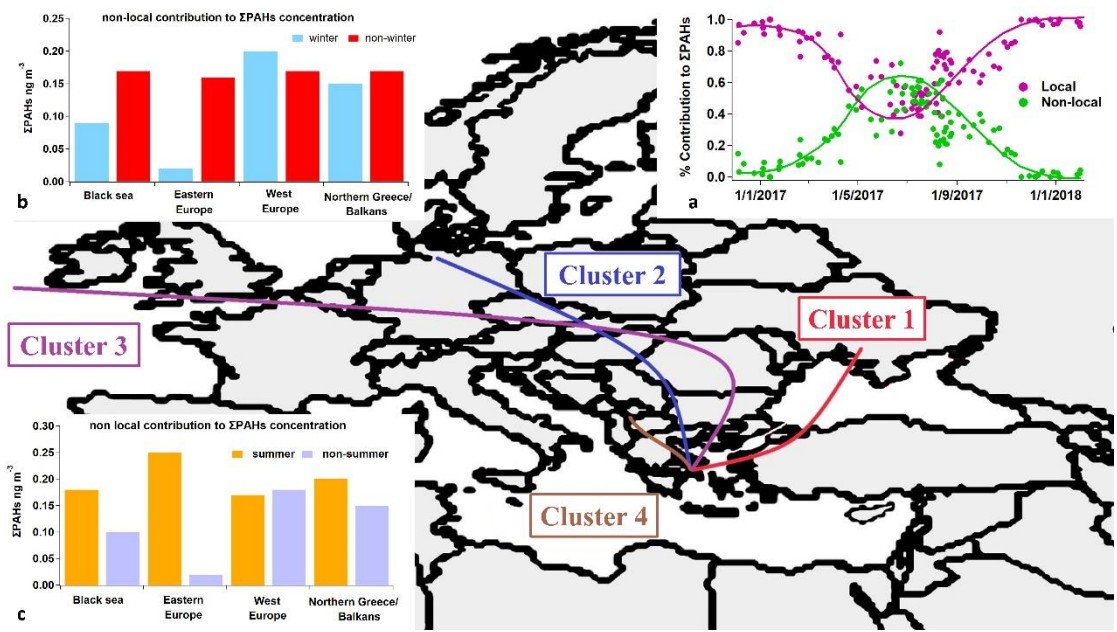

*Figure 4: The four dominant clusters of backward trajectories, Black sea (1: red), Eastern Europe (2: blue), Western Europe (3: purple) and Northern Greece /Balkans (4: brown). Panel a shows the local & non-local relative source contributions during the study period. Panels b, c show the non-local contribution to ΣPAHs concentration according to the respective air mass origin.*

When examining total PAH levels per trajectory cluster, it should be considered that relevant synoptic circulation patterns can affect the intensity of local emission sources (e.g. during wintertime cold fronts may lead to increased BB), so this impedes a fully unbiased assessment. The source apportionment results were utilized here instead, to remove the effect of local sources and to associate trajectory clusters more directly with non-local sources.

Figure 4b shows that the non-local contribution of the four clusters to ΣPAH concentrations during the non-winter period was practically the same (differences from the mean within 10%). For the winter period it can be observed that the contribution from Eastern Europe was very small compared to the other clusters. During summer (Figure 4c), the highest contributions were related to trajectories from Eastern Europe, an area identified as a hotspot of PAH production in Europe (EEA, 2018; Guerreiro et al., 2016; Rogula-Kozłowska et al., 2013). However, the small frequency (5%) of the cluster in summer should be noted. An important summertime enhancement can be also observed for non-local contributions associated with trajectories from the Black Sea area, where extensive summer agricultural burning has been identified as a significant source of carbonaceous aerosol in the Eastern Mediterranean (Amiridis et al., 2009; Sciare et al., 2005). This occurrence is mainly for the period from July to September (Sciare et al., 2008) and can be better observed using satellite imagery and fire maps (MODIS, VIIRS-NOAA). Moreover, emission inventories from GWIS (Global Wildfire Information System) verify the identified source area as a major hotspot of fire-related emissions. These fires derive predominantly from

agricultural waste burning. For example, in 2017, the fractions of total burned area that were associated with croplands were 96.5% and 96.1% in Ukraine and Turkey, respectively. Relevant information is provided in Section S5, Figure S9. For Western Europe and Northern Greece/Balkan clusters, the mean contributions remained effectively constant on a seasonal basis. In non-winter months, when non-local sources become much more prominent, contributions do not depend on air mass sector, possibly indicating that anthropogenic production of harmful aerosols is a continental-scale problem.

**3.5 Contributions to carcinogenic potency and risk assessment**

The calculated annual $BaP_{eq}$ value at the urban background site of Thissio for 2017 was 0.53 ng m$^{-3}$, higher than the one reported during 2013 for a suburban background site in the GAA that is less affected by local emissions (0.3 ng m$^{-3}$) (Alves et al., 2017). With the exception of sites in heavily industrialized areas (Kozielska et al., 2014), studies in other European and North American cities, generally calculate $BaP_{eq}$ values in the same order of magnitude as in Athens. However, much higher values, in the order of several ng m$^{-3}$ $BaP_{eq}$ are usually reported for cities in China. (Table S8). Figure S10 shows the members that contribute the most to $BaP_{eq}$ during the full measurement period; 50% of the annual $BaP_{eq}$ is attributed to BaP, consistent with studies worldwide (Amador-Muñoz et al., 2010) and confirming its importance as an indicator of PAH carcinogenic risk. The $BaP_{eq}$ value displayed a clear seasonal variability, with the highest levels during winter and the lowest in summer (Table S9). Especially during nighttime IPE in the winter of 2017-2018, the mean BaP concentration and the $BaP_{eq}$ estimate reached peak values of 2.75 ng m$^{-3}$ and 5.18 ng m$^{-3}$, respectively. Comparing to the mean $BaP_{eq}$ from all non-IPE samples (0.21 ng m$^{-3}$), it appears that these short-duration wintertime events influence disproportionately the mean annual $BaP_{eq}$ value (0.53 ng m$^{-3}$).

The PMF-resolved concentration profiles were used to obtain $BaP_{eq}$ contributions of each source. The annual contributions of biomass burning, diesel/oil and gasoline to $BaP_{eq}$ were 43%, 36% and 17%, respectively, with the remaining 4% attributed to non-local sources (Figure 3d). These results clearly highlight BB, in spite of its seasonal and highly episodic character, as a principal driver of long-term carcinogenic risk. Moreover, the carcinogenic risk of PAHs should come under appraisal in the setting of wildfires impacting with increasing frequency large urban agglomeration in vulnerable areas, such as California, Australia and Southern Europe. Such events, especially in the summer months when ambient exposure duration is increased, might become an important factor for both short- and long-term health effects in the years ahead. In the few European studies, where BB factors were apportioned and their contributions to $BaP_{eq}$ were reported (mainly in Southern European cities), these contributions were not that pronounced, even during the winter season (Callén et al., 2014; Masiol et al., 2012; Iakovides, 2021a).

Inhalation ECR values (Table S9), estimated for the annual period, were equal to $0.58 \times 10^{-6}$ (OEHHA method) and $45.73 \times 10^{-6}$ (WHO method). These estimates approached or exceeded $10^{-6}$, thought to be a threshold above which carcinogenic risks become not acceptable (EPA, 2011). At current levels, 76% of the excess risk would be attributed to wintertime exposure, mostly during IPEs, a percentage largely superior to those corresponding to the other three seasons (4-14%). Based on the stricter WHO unit risk, an excess of 4.6 cancer cases per 100,000 people can be linked to inhalation exposure to PAHs in urban

background conditions in Athens. For the 3 million living in the central Athens basin, it can be projected that the number of excess cancer cases could be well over 100.

## 4. Conclusions

Domestic biomass burning is identified as a considerable source of carcinogenic PAHs in one of the most populated regions of the Mediterranean. Overutilization of wood burning for domestic heating during the economic recession in Greece persists even today despite the improved economy, leading to a significant increase in annual urban background levels of ΣPAHs and BaP, with respect to the period preceding the recession. The local biomass burning source, which is present almost exclusively during the winter

period, emerges as the most important contributor to carcinogenic toxicity of ΣPAHs (43% on an annual basis). Therefore, wintertime exposure is seen as responsible for the largest part (76%) of the estimated excess lifetime cancer risk. This large wintertime enhancement (estimated in 2017) can be mostly attributed to a few nighttime episodes (19 events in 105 days of measurement), revealing a disproportional impact of residential BB emissions but also an opportunity for targeted intervention

measures. Given this, and the extended usage of biomass burning throughout Europe (e.g., France, Germany, Ireland and the UK), European action and policies aimed at the regulation of biomass burning emissions are immediately required as they can lead considerable benefits for public health.

Sources related to local road transport were found responsible for the major part of ΣPAH concentrations (62%), indicating that current EU planning to further curb vehicular emissions and promote

electromobility can lead to tangible results in reducing urban aerosol toxicity. However, caution is still needed for non-exhaust emissions that are also linked to enhanced particle toxicity and are expected to emerge as the principal vehicle-related particle source in the years to come (Alves et al., 2018; Daellenbach et al., 2020). In this work, there are some indications that the effects of diesel vehicles are weakening in Greek cities, which is consistent with the modernization of the commercial vehicle fleet

over the last two decades. More research in other European cities, where diesel penetration in the passenger fleets is much higher, could be rather useful to study the relative contributions and trends of diesel and gasoline vehicle contributions.

Non-local sources had a relatively small contribution to ΣPAHs level and toxicity but their relative contribution during the warm period of the year becomes comparable with that of local sources. It should

be noted that photochemistry of PAHs could be an important degradation pathway (mainly in summer) leading to formation of oxidized PAHs products, which can be considerably more toxic than parent PAHs. As this may revise the relative importance of non-local compared to local sources, oxy-PAHs are a subject of ongoing research with measurements at Thissio.

It was shown that a comprehensive observation dataset, combined with receptor modeling and back-

620 trajectory analysis provides powerful insights into the source apportionment and contributions to the health risks from PAH exposure. Despite the large body of work to date on PAHs, similar studies are surprisingly scarce in Europe and the US, so it is intended that the present study will motivate urgently needed follow-ups in other urban environments.

## Data availability

Data are available upon request, by the corresponding authors

## Author contributions

Conceptualization, AN and NM; methodology, NM, AN, IT and GG; formal analysis, IT and GG; investigation, IT, GG, AB, DP, MA, AG, CP, KT, PZ, MT, CO; writing—original draft preparation, IT and GG; writing—review and editing, IT, GG, AB, DP,NM and AN; supervision, NM and AN; funding acquisition and project administration, AN and NM.

## Competing interests

The authors declare that they have no conflict of interest.

## Acknowledgements

This work has been funded by the European Research Council, CoG-2016 project PyroTRACH (726165) H2020-EU.1.1. – Excellent. We also acknowledge support by the "PANhellenic infrastructure for Atmospheric Composition and climatE change" (MIS 5021516) which is implemented under the Action "Reinforcement of the Research and Innovation Infrastructure", funded by the Operational Programme "Competitiveness, Entrepreneurship and Innovation" (NSRF 2014-2020) and co-financed by Greece and the European Union (European Regional Development Fund). The authors thank Dr. E. Liakakou for providing BC and BC speciation data ($BC_{ff}$ and $BC_{wb}$) for Thissio, Dr. I. Hatzianestis of HCMR for providing guidance in the PAH analysis method, I. Stavroulas for the contribution in the ACSM PMF analysis, P. Kalkavouras for the trajectory cluster analysis and M. Lianou for filter collection.

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
