# Peer review of "Annual exposure to PAHs in urban environments linked to wintertime wood-burning episodes"

_Atmospheric Chemistry and Physics, 2021_

## Author Comment (AC1)

**Response to the comments of Referee # 1**

(Citation: https://doi.org/10.5194/acp-2021-393-RC1)

General comment

*This manuscript presents a database focused on polycyclic aromatic hydrocarbons (PAHs) from a year-long sampling campaign in Athens, Greece. This dataset, together with other chemical markers, was combined with receptor modeling to obtain some insights into sources and contributions to the health risks. Although there are hundreds (or even thousands) of papers published on PAHs, this combined strategy is new and has made it possible to estimate the astonishing contribution of residential biomass burning to the measured levels and estimated risks. Therefore, this study can serve as a basis for forcing policy makers to implement measures. Due to its interest for the scientific community and for stakeholders, this article deserves to be published in ACP, after review.*

We thank the referee for the assessment of our work and the constructive comments. We have revised our manuscript according to the suggestions. Please find below our responses to every point raised (in italics).

Specific comments:

*The writing of a scientific article must be impersonal. Example: "It was found that biomass burning" instead of "We find that biomass burning". Check the entire manuscript.*

This point is well taken, but it is a matter of style. That said, we will follow the suggestion and modify the manuscript accordingly.

*There are many typos throughout the manuscript. A careful review is required.*

We apologize for these oversights, the manuscript has been now carefully reviewed.

*Abstract. "responsible for annual mean PAH concentrations (31%) comparable to those from diesel/oil (33%) and gasoline (29%) sources." The sentence speaks of concentrations, but in parentheses percentages are mentioned that, I suppose, represent contributions from sources. Rephrase the sentence to make it clearer.*

Thank you for noticing this inconsistency. It is now corrected.

*Section 2.2. How was the sampling schedule? Every 3 days? One day each week?*

During the non-winter months, we analyzed two samples each week, on alternating days between weeks. We also tried to achieve a fairly representative distribution between weekday and weekend samples (69% - 31%), since it is known that this is an important temporal scale that induces variability in urban PAH levels (Dutton et al., 2010; Lough et al., 2006). The same approach was followed for the winter period, trying at the same time to analyze as many daytime-nighttime pairs as possible, to allow for the characterization of the diurnal patterns. Our intention was to fulfill and go beyond the minimum requirement of the 2004/107/EC directive for indicative measurement of PAHs (14% annual coverage, roughly 1 measurement per week, equally distributed around the year) to be able to compare with the BaP target value. These clarifications are now provided in the revised manuscript.

*Four-day air mass back trajectories, arriving at Thissio at 1000 m, were calculated. Why 1000 m?*

An altitude of 1000m was chosen to capture the regional transport of pollutants that has a high probability of affecting pollutants in the boundary layer (hence air quality and population exposure to PAHs). This altitude

is supported by numerous studies for the region studied (Grivas et al., 2018; Kalkavouras et al., 2020; Stavroulas et al., 2019). In addition according to estimates using lidar data, the planetary boundary layer over Athens extends, on average, from $892 \pm 130$ m at 00:00 UTC to $1617 \pm 324$ m at 12:00UTC (Kokkalis et al., 2020) which also supports our choice of 1000m. This discussion is included in the text.

*At what altitude is the sampling site located?*

The sampling site is located at an altitude of 146m above sea level and the inlet was at a height of 2 m above the ground. These details are now specified in the revised manuscript.

*Section 3.1. Apply statistical tests to compare means between seasons and to assess if there are significant differences and to define confidence levels.*

We applied t-tests to search for statistically significant seasonal differences (winter/ non-winter; we follow this scheme since transitional seasons in Athens are of short duration). Mean concentrations were higher during the winter period for the vast majority of PAH members, which is consistent with the strongly enhanced emissions from wintertime sources (i.e. biomass burning) and the increased volatilization and photochemical degradation of PAHs during summer. The results of the statistical analysis are now summarized in a new supplementary Table and discussed in the revised text.

*How do you explain the difference in concentrations between Dec2017/Jan2018 and Dec2016/Jan2017?*

The difference in PAHs levels between Dec2017/Jan2018 and Dec2016/Jan2017 could be explained by changes in meteorology and emission sources. For instance, BB-related primary pollutants recorded increased concentrations during the second winter, as indicated by black carbon (factor of 2.6 higher). The increased biomass burning contributions to PAH concentrations during the second winter was also verified by the results of the PMF analysis that showed higher BB contributions to $\Sigma$-PAHs by a factor of 3.8, during the second winter. Furthermore, a larger number of IPEs were observed during the second period (24 vs. 17). A modeling study (CAMx or WRF) could help identify the relative contribution of sources vs meteorology to the differences between the two winters, but this approach is out of the scope of this article.

*Partitioning of PAHs between gas and particle phases was not discussed. Concentrations of many PAHs, especially LMW species, are in reality much higher than those measured in the present study, which only include the condensed form.*

Excellent point. Previous works conducted in Athens  (Mandalakis et al., 2002; Sitaras and Siskos, 2001, 2008; Vasilakos et al., 2007), that performed both gas- and particle-phase measurements, reported that LMW PAHs were found predominantly in the gas phase (gas-phase fractions: 88-97%), MMW PAHs in both gas and particulate phase (gas phase: 46-90%) and HMW PAHs mainly in the particle phase (particle phase: 71-100%).

By applying the methodology developed by Pankow (1994) we found that the largest fractions of LMW PAHs are partitioned in the gas phase (71% - 100%). Most MMW and all HMW PAHs are partitioned preferentially in the particle phase, at rates ranging between 90%-100%, with the exception of the lighter Flt and Pyr. The results support the decision not to include the LMW PAH members in the source apportionment analysis, since their temperature-dependent partitioning could induce large uncertainties in the source-receptor relationships. It also appears that it was indeed useful to downweight the involvement of Flt and Pyr (15%, 25% in the particle phase) in the PMF model. Moreover, it becomes obvious that caution is needed in the interpretation of results for LMW members in PAH studies that measure only the particle phase, especially using diagnostic ratios, since their temperature-dependent concentrations aren't sufficiently representative of the relationships between PAH members as emitted from sources.

The methodology and outcomes are presented as supplementary material and the main outcomes are discussed in the revised text.

*It should also be borne in mind that PAHs suffer photochemical reactions leading to the formation of nitro-and oxy-derivatives.*

We would like to thank the reviewer for this suggestion. Note that we are currently working on the analysis of nitro and oxy-derivatives of parent PAHs at the same site, and we will soon be able to verify and provide also a quantitative assessment of secondary photochemical processes (this is now indicated in the conclusions sections as a proposed research direction).

*Authors should look for correlations with ozone and NOx concentrations provided by the air quality measurement network to better interpret the results obtained.*

This is a good point. In-situ measurements of $O_3$, $NO_x$ and CO were available at the measurement site, using reference-grade instruments. We also obtained $NO_x$ from a nearby, traffic-impacted, official station (Athinas Str.) of the AQ monitoring network, to see whether there are traffic-background contrasts in the correlations. We performed the correlation analysis separately for winter and non-winter months. The results are presented in a new supplementary table and discussed in the text.

*End of page 12. Non-local contributions were associated with trajectories from the Black Sea area, where "extensive summer agricultural burning has been identified". Legislation within the EU has largely outlawed the practice of field burning agricultural wastes, especially in summer. Authors must make sure that it is agricultural burning or wildfires. For this purpose, fire maps and emission inventories by EMEP/EEA (which include emissions from agricultural burning) should be consulted.*

We would like to thank the reviewer for his/her comment. Although the (EU) 1306/2013 regulation has forbidden the practice of field burning of agricultural wastes, large scale agricultural biomass burning is still being observed in countries of the Black Sea region, that are not directly bound by EU legislation like Ukraine, Russia (southwestern oblasts) and countries in the Caucasus area (Hall et al., 2021; McCarty et al., 2017). This phenomenon is specifically intensified the period from July to September (Sciare et al., 2008). We have examined GWIS (Global Wildfire Information System) emission inventories and also satellite fire maps (TERRA and AQUA MODIS, VIIRS-NOOA), that verify the identified source area as a major hotspot of fire-related emissions. We provide indicative supplementary figures and data in the Supplement. Discerning between wildfires and agricultural burning can be difficult. However, we examined land use/cover maps in the region, superimposed over the fire maps and it appears that the majority of identified fires occurred in the extensive croplands of Ukraine and Southern Russia. Indicative figures are also included in the Supplement.

*Conclusions. I'm not sure about the promotion of electromobility. While exhaust emissions would decrease, non-exhaust emissions would, on the contrary, increase. Heavier battery electric vehicles may result in more tyre/brake wear and resuspension emissions than the current vehicle fleet. Several recent studies have shown that non-exhaust emissions are at least as toxic as those from exhaust emissions. One thing is for sure ... The use of public transport instead of individual transport should be promoted.*

Thank you for bringing up this point. EU seems bound to the direction of electromobility, having recently proposed a ban on fossil-fuel powered cars from 2035, a decision that will lead to large cutbacks of vehicular exhaust emissions. However, in the next decades, the majority of PM from the road transport sector is expected to derive from non-exhaust emissions, even without the enforcement of electromobility (Daellenbach et al., 2020). This complex tradeoff between exhaust and non-exhaust emissions is now described in the conclusions, along with the necessity to adopt measures and practices for the reduction of non-exhaust vehicular emissions as that suggested by the reviewer. We also note, that the use of heavier private vehicles could leaded to increased traffic-induced resuspension of particles containing PAHs (Alves et al., 2018; Oliveira et al., 2011).

**References**

Alves, C. A., Evtyugina, M., Vicente, A. M. P., Vicente, E. D., Nunes, T. V., Silva, P. M. A., Duarte, M. A. C., Pio, C. A., Amato, F. and Querol, X.: Chemical profiling of PM10 from urban road dust, Sci. Total Environ., 634, 41–51, doi:10.1016/j.scitotenv.2018.03.338, 2018.

Daellenbach, K. R., Uzu, G., Jiang, J., Cassagnes, L.-E., Leni, Z., Vlachou, A., Stefenelli, G., Canonaco, F., Weber, S., Segers, A., Kuenen, J. J. P., Schaap, M., Favez, O., Albinet, A., Aksoyoglu, S., Dommen, J., Baltensperger, U., Geiser, M., El Haddad, I., Jaffrezo, J.-L. and Prévôt, A. S. H.: Sources of particulate-matter air pollution and its oxidative potential in Europe, Nature, 587(7834), 414–419, doi:10.1038/s41586-020-2902-8, 2020.

Dutton, S. J., Rajagopalan, B., Vedal, S. and Hannigan, M. P.: Temporal patterns in daily measurements of inorganic and organic speciated PM2.5 in Denver, Atmos. Environ., 44(7), 987–998, doi:10.1016/j.atmosenv.2009.06.006, 2010.

Grivas, G., Cheristanidis, S., Chaloulakou, A., Koutrakis, P. and Mihalopoulos, N.: Elemental Composition and Source Apportionment of Fine and Coarse Particles at Traffic and Urban Background Locations in Athens, Greece, Aerosol Air Qual. Res., 18(7), 1642–1659, doi:10.4209/aaqr.2017.12.0567, 2018.

Hall, J. V, Zibtsev, S. V, Giglio, L., Skakun, S., Myroniuk, V., Zhuravel, O., Goldammer, J. G. and Kussul, N.: Environmental and political implications of underestimated cropland burning in Ukraine, Environ. Res. Lett., 16(6), 064019, doi:10.1088/1748-9326/abfc04, 2021.

Kalkavouras, P., BougiatiotI, A., Hussein, T., Kalivitis, N., Stavroulas, I., Michalopoulos, P. and Mihalopoulos, N.: Regional New Particle Formation over the Eastern Mediterranean and Middle East, Atmosphere (Basel)., 12(1), 13, doi:10.3390/atmos12010013, 2020.

Kokkalis, P., Alexiou, D., Papayannis, A., Rocadenbosch, F., Soupiona, O., Raptis, P.-I., Mylonaki, M., Tzanis, C. G. and Christodoulakis, J.: Application and Testing of the Extended-Kalman-Filtering Technique for Determining the Planetary Boundary-Layer Height over Athens, Greece, Boundary-Layer Meteorol., 176(1), 125–147, doi:10.1007/s10546-020-00514-z, 2020.

Lough, G. C., Schauer, J. J. and Lawson, D. R.: Day-of-week trends in carbonaceous aerosol composition in the urban atmosphere, Atmos. Environ., 40(22), 4137–4149, doi:10.1016/j.atmosenv.2006.03.009, 2006.

Mandalakis, M., Tsapakis, M., Tsoga, A. and Stephanou, E. G.: Gas–particle concentrations and distribution of aliphatic hydrocarbons, PAHs, PCBs and PCDD/Fs in the atmosphere of Athens (Greece), Atmos. Environ., 36(25), 4023–4035, doi:10.1016/S1352-2310(02)00362-X, 2002.

McCarty, J. L., Krylov, A., Prishchepov, A. V, Banach, D. M., Tyukavina, A., Potapov, P. and Turubanova, S.: Agricultural Fires in European Russia, Belarus, and Lithuania and Their Impact on Air Quality, 2002–2012, in Land-Cover and Land-Use Changes in Eastern Europe after the Collapse of the Soviet Union in 1991, edited by G. Gutman and V. Radeloff, pp. 193–221, Springer International Publishing, Cham., 2017.

Oliveira, C., Martins, N., Tavares, J., Pio, C., Cerqueira, M., Matos, M., Silva, H., Oliveira, C. and Camões, F.: Size distribution of polycyclic aromatic hydrocarbons in a roadway tunnel in Lisbon, Portugal, Chemosphere, 83(11), 1588–1596, doi:10.1016/j.chemosphere.2011.01.011, 2011.

Pankow, J. F.: An absorption model of the gas/aerosol partitioning involved in the formation of secondary organic aerosol, Atmos. Environ., 28(2), 189–193, doi:10.1016/1352-2310(94)90094-9, 1994.

Sciare, J., Oikonomou, K., Favez, O., Liakakou, E., Markaki, Z., Cachier, H. and Mihalopoulos, N.: Long-term measurements of carbonaceous aerosols in the Eastern Mediterranean: evidence of long-range transport of biomass burning, Atmos. Chem. Phys., 8(18), 5551–5563, doi:10.5194/acp-8-5551-2008, 2008.

Sitaras, I. E. and Siskos, P. A.: Levels of Volatile Polycyclic Aromatic Hydrocarbons in the Atmosphere of Athens, Greece, Polycycl. Aromat. Compd., 18(4), 451–467, doi:10.1080/10406630108233820, 2001.

Sitaras, I. E. and Siskos, P. A.: The role of primary and secondary air pollutants in atmospheric pollution: Athens urban area as a case study, Environ. Chem. Lett., 6(2), 59–69, doi:10.1007/s10311-007-0123-0, 2008.

Stavroulas, I., Bougiatioti, A., Grivas, G., Paraskevopoulou, D., Tsagkaraki, M., Zarmpas, P., Liakakou, E., Gerasopoulos, E. and Mihalopoulos, N.: Sources and processes that control the submicron organic aerosol composition in an urban Mediterranean environment (Athens): a high temporal-resolution chemical composition measurement study, Atmos. Chem. Phys., 19(2), 901–919, doi:10.5194/acp-19-901-2019, 2019.

Vasilakos, C., Levi, N., Maggos, T., Hatzianestis, J., Michopoulos, J. and Helmis, C.: Gas–particle concentration and characterization of sources of PAHs in the atmosphere of a suburban area in Athens, Greece, J. Hazard. Mater., 140(1–2), 45–51, doi:10.1016/j.jhazmat.2006.06.047, 2007.

---

## Author Comment (AC2)

**Response to the comments of Referee # 2**

(Citation report: https://doi.org/10.5194/acp-2021-393-RC2)

**General comments**

*This study investigated the sources of ambient carbonaceous aerosol, particularly PAHs, in the area of Athens, Greece. The authors made good use of filter-based measurement methods for PMF analysis with organic molecular markers to link PAH concentrations and composition with specific sources, e.g. biomass burning. The analysis is thorough and generally clear. A few issues should be addressed: (a) figure quality could be improved,(b) the authors should more clearly state the novelty of the study relative to previous studies in the region, and (c) ACSM data should be discussed in more detail and in the context of the filter samples.*

We thank the reviewer for the positive assessment of our work. We have carefully considered the general and specific suggestions and incorporated them in the revised manuscript. Please find below our detailed response to the reviewer's comments.

**Specific comments**

*All figures: The figures are pixelated and hard to see/read – please save the images in higher resolution*

We believe that the pixelation was due to the online production of the pdf file. All figures have been now produced as high resolution image files and can be submitted separately as well. A better color selection was also made in the map figure (Fig. 4).

*Seasonal trends: How does Dec/Jan 2016-2017 compare with Dec/Jan 2017-2018 in terms of PAH concentrations, source contributions, etc.?*

This point was also raised by the reviewer 1 and addressed accordingly.

*Lines 74-84: There have been several previous studies of ambient air quality in Greece and how it is impacted by domestic biomass burning in particular, as the authors reference earlier in the introduction. Please explicitly state the novelty of the present study: other than being more recent, how do the sampling techniques/locations/times and analyses employed provide new insight? This would also be a good opportunity to discuss the use of complementary filter-based and online (?) techniques and what additional insight these provide, as using both of these is a strength of the study.*

Thank you for providing the opportunity to describe more clearly the novel aspects of our work. The elements of originality of our work are now presented in detail in the last paragraph of the introduction.

*Lines 128-133: Describe ACSM monitoring in more detail. Was this conducted at the same site as the filter sampling? What were the sampling dates/duration?*

More details on ACSM measurements (operating principle, QA/QC, PMF methodology etc.) are provided in the revised manuscript.

*Lines 133-136: Describe the method used to apportion BC into fossil fuel and wood burning contributions, as this is a secondary calculation for interpretation rather than primary data from the instrument*
The Aethalometer model (Drinovec et al., 2015; Sandradewi et al., 2008) that is used internally by the AE33 7-$\lambda$ aethalometer to apportion BC into source-specific components, is now briefly described.

*Lines 180-183: Include numbers from this study for reference, and change "which however included 7" to "7 of which were"*
We included indicative numbers and also discussed that the cited study reported street-site concentrations higher by 44 and 55% compared to urban background sites, for $\Sigma$-PAHs and BaP, respectively.

*Figure 1: Stacked bar plots with this many different categories/colors do not provide useful information. I recommend presenting the individual PAH concentrations in a table, which would facilitate direct comparisons with other data sets in the future. The barplot could instead show monthly averages of HMW/MMW/LMW PAHs using the same colors as in the pie charts below, which would provide the reader with an overview of both absolute and relative concentrations of general PAH categories.*
The figure was redesigned according to the suggestions of the reviewer.

*Lines 234-242: Why is the ACSM data only discussed here? If it is included in the manuscript, it should be discussed further: for example, how the ACSM SoFi and filterbased PMF source apportionment compare, and the temporal trends in ACSM data. Which ACSM sources would likely include PAHs based on chemical signatures, and do these source contributions correspond to PAH concentrations? Also, clarify exactly what m/z 60 and 73 are thought to represent (for readers less familiar with AMS data). Do they correspond with filter-based levoglucosan measurements?*
Regarding the potential inclusion of PAHs in ACSM measurements based on the mass spectrometric chemical signatures, that is suggested by the reviewer, we note that the upper *m/z* peak that is detected by our Q-ACSM is *m/z* 140 and therefore it wasn't possible to associate specific fragments with PAHs (naphthalene at *m/z* 128 is used by the ACSM as an internal standard to monitor detector performance) (Ng et al., 2011). The identification of distinctive PAH *m/z* signals in mass spectra has been reported by measurements with more advanced instrumentation, such as the ToF-ACSM (Zheng et al., 2020) and aerosol mass spectrometers (AMS) (Dzepina et al., 2007; Herring et al., 2015).
However, following his/her suggestion, we included in the revised manuscript additional ACSM data and performed correlation analysis with the source contributions from filter-based PMF, to explore commonalities in the temporal variabilities of the two approaches. The above analysis was incorporated in the revised manuscript.

*Lines 293-295: How was CO measured?*

CO was measured using a reference-grade NDIR monitor (APMA 360, Horiba Inc.). All details regarding measurement of regulatory pollutants are now included in the last paragraph of section 2.2.

*Figure 3: Given that the carcinogenic risk of PAHs is summarized in the analysis as BaP equivalents, what is the purpose of showing both "c" and "d" pies (carcinogenic PAHs and BaPeq)? Though individual PAHs have differing degrees of toxicity, the numbers end up being very similar between c and d, so it doesn't seem necessary to present both in the main manuscript. Individual pies for OC and EC apportionment would be more interesting from a chemical perspective. It would also be helpful to label each pie with a short title, in addition to the more detailed descriptions in the figure legend.*

The Figure was amended according to the suggestion of the reviewer, omitting panel (c). Pie charts for source contributions to OC, EC were included in the supplement (since these are already shown in the source profiles of Figure 2).

**Technical corrections**

*Line 26: Change "effective" to "present"*

Amended.

*Line 27: Change "lead" to "leads"*

Amended.

*Line 32: This refers to total measured PAHs? Clarify*

It refers to the ensemble of the 12 PAHs included in the PMF analysis. It is now specified.

*Line 41: Add "such as" before "power and industrial plants"*

Amended.

*Line 46: Change "Particular PAH members" to "Several PAHs"*

Amended.

*Line 49: BaP equivalents? Clarify.*

While the directive highlights the necessity of measurements for additional PAHs, the set target value refers only to BaP, which is selected as a marker for the carcinogenic risk of polycyclic aromatic hydrocarbons in ambient air. The part was clarified.

*Lines 54-56: Wouldn't it make sense to cite Saffari et al 2013 here, too?*

We agree, this was indeed the first publication to directly address the RWB issue in Greek cities. It is now included.

*Lines 178-180: Need references*

We added references to (Andreou and Rapsomanikis, 2009; Mantis et al., 2005) that measured at traffic sites and also to (Pateraki et al., 2019) where the urban industrial site recording the highest concentrations (Table S3) is in fact close to a major avenue in the center of Athens.

*Lines 329-330: Change "time-series" to "from non-local sources" (I believe that is what this is referring to, but currently it is unclear)*

Amended.

*Line 336: change "lighter members" to "LMW PAHs"*

Amended.

*Lines 382-383: "transport" should be followed by a semicolon (;), not a comma (,) and "its variability and origin" should be "their variability and origins"*

Amended.

*Line 414: Shouldn't BaPeq be written with a subscript, i.e. BaPeq? (here and throughout the rest of the manuscript)*

Amended.

*Line 441: Do "GAA" and "central Athens basin" refer to the same region? If not, clarify, and if so, be consistent*

The GAA includes areas outside of the central Athens basin, namely the Thriassion Plain to the west and the Mesogeia Plain to the east. That is why we mention in section 2 that it has a larger population (3.8 million), compared to the 3 million stated in Line 441. We chose to make the projection just for the population of the Athens basin because we can't claim that levels at Thissio are representative of conditions in the Thriassion (industrial area) or Mesogeia (mostly rural background conditions).

**References**

Andreou, G. and Rapsomanikis, S.: Polycyclic aromatic hydrocarbons and their oxygenated derivatives in the urban atmosphere of Athens, J. Hazard. Mater., 172(1), 363–373, doi:10.1016/j.jhazmat.2009.07.023, 2009.

Drinovec, L., Močnik, G., Zotter, P., Prévôt, A. S. H., Ruckstuhl, C., Coz, E., Rupakheti, M., Sciare, J., Müller, T., Wiedensohler, A. and Hansen, A. D. A.: The "dual-spot" Aethalometer: An improved measurement of aerosol black carbon with real-time loading compensation, Atmos. Meas. Tech., 8(5), 1965–1979, doi:10.5194/amt-8-1965-2015, 2015.

Dzepina, K., Arey, J., Marr, L. C., Worsnop, D. R., Salcedo, D., Zhang, Q., Onasch, T. B., Molina, L. T., Molina, M. J. and Jimenez, J. L.: Detection of particle-phase polycyclic aromatic hydrocarbons in Mexico City using an aerosol mass spectrometer, Int. J. Mass Spectrom., 263(2–3), 152–170, doi:10.1016/j.ijms.2007.01.010, 2007.

Herring, C. L., Faiola, C. L., Massoli, P., Sueper, D., Erickson, M. H., McDonald, J. D., Simpson, C. D., Yost, M. G., Jobson, B. T. and VanReken, T. M.: New Methodology for Quantifying Polycyclic Aromatic Hydrocarbons (PAHs) Using High-Resolution Aerosol Mass Spectrometry, Aerosol Sci. Technol., 49(11), 1131–1148, doi:10.1080/02786826.2015.1101050, 2015.

Mantis, J., Chaloulakou, A. and Samara, C.: PM10-bound polycyclic aromatic hydrocarbons (PAHs) in the Greater Area of Athens, Greece, Chemosphere, 59(5), 593–604, doi:10.1016/j.chemosphere.2004.10.019,

2005.

Ng, Y. L., Mann, V. and Gulabivala, K.: A prospective study of the factors affecting outcomes of nonsurgical root canal treatment: Part 1: Periapical health, Int. Endod. J., 44(7), 583–609, doi:10.1111/j.1365-2591.2011.01872.x, 2011.

Pateraki, S., Fameli, K.-M., Assimakopoulos, V., Bougiatioti, A., Maggos, T. and Mihalopoulos, N.: Levels, Sources and Health Risk of PM2.5 and PM1-Bound PAHs across the Greater Athens Area: The Role of the Type of Environment and the Meteorology, Atmosphere (Basel)., 10(10), 622, doi:10.3390/atmos10100622, 2019.

Sandradewi, J., Prévôt, A. S. H., Szidat, S., Perron, N., Alfarra, M. R., Lanz, V. A., Weingartner, E. and Baltensperger, U.: Using Aerosol Light Absorption Measurements for the Quantitative Determination of Wood Burning and Traffic Emission Contributions to Particulate Matter, Environ. Sci. Technol., 42(9), 3316–3323, doi:10.1021/es702253m, 2008.

Zheng, Y., Cheng, X., Liao, K., Li, Y., Li, Y. J., Huang, R.-J., Hu, W., Liu, Y., Zhu, T., Chen, S., Zeng, L., Worsnop, D. R. and Chen, Q.: Characterization of anthropogenic organic aerosols by TOF-ACSM with the new capture vaporizer, Atmos. Meas. Tech., 13(5), 2457–2472, doi:10.5194/amt-13-2457-2020, 2020.